# Current evidence and future direction on evaluating the anticancer effects of curcumin, gingerols, and shogaols in cervical cancer: A systematic review

Unwaniah Abdull Rahim[1], Marami Mustapa[2], Nik Noorul Shakira Mohamed Shakrin[3,4], Armania Nurdin[5,6]*, Nursiati Mohamad Taridi[1], Yasmin Anum Mohd Yusof[1], Mariam Firdhaus Mad Nordin[7], Nur Aishah Che Roos[8]*

1 Biochemistry Unit, Faculty of Medicine and Defence Health, National Defence University of Malaysia, Kuala Lumpur, Malaysia, 2 Anatomy Unit, Faculty of Medicine and Defence Health, National Defence University of Malaysia, Kuala Lumpur, Malaysia, 3 Centre for Tropicalization (CENTROP), National Defence University of Malaysia, Kuala Lumpur, Malaysia, 4 Medical Microbiology and Immunology Unit, Faculty of Medicine and Defence Health, National Defence University of Malaysia, Kuala Lumpur, Malaysia, 5 Department of Biomedical Science, Faculty of Medicine and Health Science, Universiti Putra Malaysia, Serdang, Selangor, Malaysia, 6 Laboratory of UPM-MAKNA Cancer Research (CANRES), Institute of Bioscience, Universiti Putra Malaysia, Serdang, Selangor, Malaysia, 7 Malaysia-Japan International Institute of Technology, UTM Kuala Lumpur, Jalan Sultan Yahya Petra, Kuala Lumpur, Malaysia, 8 Pharmacology Unit, Faculty of Medicine and Defence Health, National Defence University of Malaysia, Kuala Lumpur, Malaysia

* nuraishah@upnm.edu.my (NACR); armania@upm.edu.my (AN)

**Data Availability Statement:** All relevant data are within the manuscript and its Supporting

## Abstract

Cervical cancer ranked fourth most common malignancy among women worldwide despite the establishment of vaccination programmes. This systematic review evaluates the anticancer properties of turmeric and ginger bioactive compounds, specifically curcumin, 6/10-gingerol, and 6/10-shogaol, and their combination in cervical cancer through *in-vitro* and *in-vivo* models. A comprehensive electronic search was performed using Science Direct, PubMed, and Scopus from inception until the second week of June 2024 for studies published in English. Only studies investigating the effects of curcumin, gingerol, shogaol, and/ or their combination in human cervical cancer cell lines and/or rodent animal models implanted with cervical cancer xenografts were included. Altogether, 27 studies were included in this review. The evidence gathered indicated that curcumin, 6/10-gingerol and 6-shogaol exert their anticancer action through modulation of cell signalling pathways, including AMPK, WNT, PI3K/AKT, and NF-κB pathway, and mediators including Bax/Bcl2, TNF-α, EGFR, COX-2, caspases-3, -9, p53, and pRb. However, the synergistic effect of these bioactive compounds is not known due to lack of evidence. In conclusion, curcumin, 6/10-gingerols, and 6-shogaols hold promise as therapeutic agents for cervical cancer. Yet, further research is essential to understand their combined efficacy, emphasising the need for additional studies exploring the synergistic anticancer effects of these bioactive compounds. Additional factors to explore include long-term effects and susceptibility of chemoresistant cervical cancer cells towards curcumin, shogaols, and gingerols.

Information files. All raw data required to replicate the results of this study are either reported in the manuscript or compiled in the Supporting Information documents. The data on screening and selection process of the studies are available from the Zenodo repository at https://zenodo.org/records/14000521.

**Funding:** The publication of this work is supported financially by 1. The Fundamental Research Grant Scheme (FRGS) Ministry of Higher Education (MoHE) Malaysia (FRGS/1/2021/SKK0/UPNM/02/2) - Dr Nik Noorul Shakira Mohammed Shakrin 2. National Defence University of Malaysia (NDUM) - Dr Nur Aishah Che Roos 3. Universiti Putra Malaysia (UPM) -Dr Armania Nurdin The funders do not play any role in the study design, data collection and analysis, decision to publish, or preparation of the manuscript.

## Introduction

In spite of the vaccination programme establishment, cervical cancer remains a significant global health issue, ranking as the fourth most common malignancy among women worldwide, leading to a high fatality rate [1]. In 2020 alone, there were over 604 127 cases and 341,831 deaths attributed to cervical cancer [1]. The development of cervical cancer is associated with various risk factors, including high-risk human papillomavirus (HPV) infection transmitted through sexual intercourse, age, smoking, high number of childbirths, long-term oral contraceptive usage, and diet [2]. Among these risk factors, persistent HPV infection is considered the primary factor for cervical carcinomas [2].

The prevalence of HPV subtypes varies geographically. In Europe, HPV subtypes 16, 18 and 45 are predominant [3]. In Bahamas and Brazil, subtypes 16 and 18 are more common [4, 5]. Moving to the Asian region, China shows the dominance of subtypes 16, 58 and 33 [6], while India predominantly reports subtypes 16 and 18, which collectively account for 85% of cases [7], and Malaysia exhibits a higher prevalence of subtypes 52 and 66 [8]. It is noteworthy that HPV subtypes 16 and 18 account for more than 70% of cervical cancer worldwide [7, 9–11]. HPV is a known cause of cancer as it interferes with the normal function of two important tumour suppressor proteins, p53 and pRb [12]. The oncoproteins E6 and E7 produced by high-risk types of HPV interact with these proteins, disrupting their regular activity and leading to uncontrolled cell growth and the development of cancer [12].

Given the global impact of cervical cancer, extensive research has been conducted to develop effective treatments at different stages of the cancer. The most widely used treatment modalities for cervical cancer involve surgery, radiotherapy, and chemotherapy [13]. Chemotherapy, in particular, uses low molecular weight drugs to target and destroy tumour cells or, at the very least, inhibit their growth [13]. Different types of chemotherapy drugs are used, including anti-metabolites (e.g., 5-fluorouracil, methotrexate), DNA-interactive agents (e.g., cisplatin, doxorubicin), and anti-tubulin agents (e.g., taxanes) [14]. However, chemotherapy and radiation therapy often lead to various adverse effects and toxicity, resulting in damage to non-targeted tissues, such as hair loss, neurotoxicity, nausea, anaemia, and neutropenia since they act on both tumour cells and healthy cells. These undesirable effects may compromise treatment efficacy and negatively impact the quality of life of cancer patients [14, 15]. Recent advancements in CART-T cell therapy in cancer treatment have demonstrated efficacy in various cancers where the immune cells are genetically designed to better recognise and enhance the immune response to eliminate the cancer cells [16]. However, its effectiveness in solid tumours remains limited due to the immunosuppressive tumour microenvironment, which represents significant challenges [17].

To address the limitations of current treatment, there is a pressing need to develop new anti-cancer drugs that are more effective and have fewer adverse effects. In recent years, there has been growing interest in the use of plant-derived compounds as complementary or alternative therapies for cancer treatment. In fact, studies have shown that 50 to 60% of cancer patients in the United States used plant-derived supplements alongside conventional chemotherapy and/or radiation therapy [14]. These natural compounds are preferred due to their potential for minimal side effects, high efficacy, low cost and easier accessibility [15]. Bioactive compounds derived from various plants, such as turmeric and ginger of the Zingiberaceae family, have demonstrated anti-cancer properties in both *in-vitro* and *in-vivo* studies [18].

Turmeric (*Curcuma longa*), a spice commonly used in Southeast Asian cuisine, contains a bioactive compound called curcumin, which has been extensively studied for its therapeutic properties [19]. Curcumin is a polyphenol and has been found to be the most potent among curcuminoids contained in turmeric. Curcumin is recognised for its therapeutic effects on

several diseases, such as cancer, autoimmune disease, and various inflammatory conditions [20]. This is primarily due to its ability to modulate immune responses, making it beneficial in inducing inflammation and regulating immune functions. Recently, it has been shown to exhibit anti-ageing properties by influencing molecular pathways to ageing, such as AMPK, and the inhibition of NF-κB and mTOR, which contributes to delaying age-associated disorders [21].

Numerous laboratory studies have demonstrated that curcumin has a great capacity to block various biochemical processes involved in cancer survival and growth [22, 23]. It can regulate cell proliferation, angiogenesis, metastasis, apoptosis, cancer-associated inflammation, and drug resistance by directly or indirectly binding to molecular targets [24, 25]. Curcumin has also been studied in combination with other natural or synthetic compounds to overcome the limitations of chemotherapy while reducing adverse effects. In response to growing studies on curcumin, several clinical trials have shown that oral administration of curcumin is safe, well-tolerated, and has an acceptable blood chemistry profile with no significant toxicity [19, 26].

Ginger (*Zingiber officinale*) is another widely used spice which comes in numerous forms and is considered part of the 'holy trinity' in Chinese cuisine. Also, it has been recognised for its potential chemopreventive properties [27]. Gingerols and shogaols are natural compounds found in ginger that contribute to its odour and flavour [28]. These compounds have a high oral bioavailability, meaning they are easily absorbed and utilised by the body when consumed as part of the diet. Gingerols exist in different forms, including 4-gingerol, 6-gingerol, 8-gingerol, and 10-gingerol [29, 30]. Whereas shogaols exist in the form of 4-shogaol, 6-shogaol, 8-shogaol, 10-shogaol, and 12-shogaol. Ginger and its bioactive compounds have been shown to possess various biological activities, such as anti-cancer, anti-inflammatory, antioxidant, anti-microbial, and anti-allergic properties [31].

Many experimental studies have investigated the chemopreventive properties of curcumin, gingerol, and shogaol in different types of malignancies including colon cancer [32], bone cancer [33, 34] and breast cancer [23, 35]. However, only a limited number of studies explored the anti-cancer effects of the selected compounds on cervical cancer. Furthermore, no existing review explored the anti-cancer effect of these bioactive compounds when used in combination. By recognising the current gap in experimental studies, this systematic review aims to make a significant contribution to the field by focusing on collating and critically assessing *in-vitro* and *in-vivo* studies evaluating the anti-cancer effects of curcumin, gingerol, shogaol, and/or their combination for cervical cancer treatments. The primary objective is to systematically appraise and synthesise preclinical data, providing a multifaceted understanding of curcumin, gingerol, shogaol, and/or their combination in exerting their anti-cancer effect in cervical cancer, particularly in identifying the molecular mechanism and pathway being targeted. By integrating findings from *in-vitro* and *in-vivo* studies, this review is able to capture comprehensive insights into their chemotherapeutic potential with a more detailed understanding of the underlying mechanisms and pathways that these compounds of interest affect, regardless of the preparation of the compounds. The exclusion of clinical studies in evaluating the anti-cancer effects of curcumin, gingerol and shogaol is justified, as overall human studies primarily focus on efficacy and adverse reactions rather than elucidating the specific molecular targets involved.

## Material and methods

The development of the systematic review protocol is in accordance with the Preferred Reporting Items for Systematic review and Meta-Analysis (PRISMA) [36, 37]. The protocol for this review is registered on the International Prospective Register of Systematic Reviews (PROSPERO ID: CRD42022334940) and is published elsewhere [38]. The search period for this

review is slightly extended to a more recent date compared to the information provided in the registered and published protocol for comprehensive current evidence.

## Search strategy and sources

A search strategy using a combination of Medical Subject headings (MeSH) and keywords together with Boolean operators was developed as follows: (Curcumin OR turmeric OR Curcuma OR ginger OR gingerol OR Zingiber OR *Z.officinale* OR shogaol) AND ((Cervical OR cervix OR "human papillomavirus") AND (cancer OR carcinoma OR malignancy OR tumour)) AND (HeLa OR SiHa OR CaSki OR "cell line" OR "*in-vitro*" OR animal OR rodent OR "*in-vivo*").

Electronic databases, including PubMed, Scopus, and Science Direct, were searched for eligible studies from inception until the second week of June 2024. The last search was conducted on 14 June 2024. The references of eligible articles were also screened for relevant studies. Only English publications were considered for this review. Grey literature or evidence not published in academic publications were excluded.

## Inclusion and exclusion criteria

The inclusion criteria based on the PICOS framework are as follows: (1) Population: human cervical cancer cell lines (e.g., HeLa, SiHa, CaSki) and tumour-bearing animals implanted with human cervical cancer xenografts. The animal model used was restricted to only the rodent family. All cervical cell lines and animal models were included regardless of the presence or absence of HPV infection; (2) Intervention and comparators: single and/or combined bioactive compounds comprising gingerol, shogaol, and curcumin regardless of dose and duration of the intervention were eligible for inclusion. The studied bioactive compounds may be prepared in any form. The bioactive compounds, as stated earlier, may be compared with each other, against standard cervical cancer chemotherapy as a positive control (e.g., cisplatin, paclitaxel, 5-fluorouracil) or with cells/animals that were not treated (negative control); (3) Outcome measured: The primary outcome include anti-cancer activity and cytotoxic effects of the bioactive compounds assessed through cell viability, cell cycle growth, cell apoptosis, protein expression, gene expression activity, tumour size, or histological changes via standard procedures. The secondary outcome includes the signalling pathway(s) and molecular target(s) involved in the anti-cancer effect of the studied bioactive compounds; and (4) Study design: controlled *in-vitro* studies involving human cervical cancer cell lines and/or controlled *in-vivo* studies using tumour-bearing animals implanted with human cervical cancer xenografts.

Exclusion criteria include: (1) Population: preclinical studies using non-cervical cancer cell lines and other types of animal models; (2) Intervention and comparator: combination therapy with another bioactive compound not as listed in the inclusion criteria and/or drugs, bioactive compounds delivery aided by external biological factors, e.g., nanocarriers; and biologically enhanced or conjugated bioactive compounds. (3) Studied bioactive compounds compared to anti-cancer drugs not conventionally used for cervical cancer were excluded. (4) Studies that combined the effects of studied bioactive compounds with another standard treatment modality, such as radiation therapy, were also excluded, and (5) Studies enrolling human subjects were excluded. Reviews, editorials, conference proceedings, and abstracts where the full texts were unobtainable were excluded.

## Study screening and selection

Citations obtained from the electronic database search were compiled into a web-based app known as Rayyan [39], a semi-automated artificial intelligence tool for the identification of

studies in conducting a systematic review. Two independent reviewers (UAR and NA) screened the titles and abstracts retrieved for the identification of eligible studies. Any discrepancy was solved by discussion between the two reviewers or by consulting a third reviewer (YA) if necessary. The full text of the eligible studies was screened by two independent reviewers (UAR and NA), and only studies that met the inclusion criteria were included.

## Data management and extraction

Two independent reviewers (UAR and NA) created and piloted a standardised data extraction form for the extraction of variables. The following information was extracted for included study characteristics: Name of first author, year of publication, country, and study design. Based on the PICOS framework, the following information was extracted: cervical cancer cell line used, animal model including sex of rodent and type of tumour cell, sample size, type of bioactive compound used and its comparator including dose and frequency/duration. The following information was extracted for the primary outcomes: half maximal inhibitory concentration ($IC_{50}$) of bioactive compound used, cell viability, cell cycle growth, cell apoptosis, protein expression, gene expression activity, volume of tumour, and/or histological changes via standard procedures. Additionally, information on the mechanism of signalling pathway(s) and molecular target(s) studied was extracted where available. Any disagreement between the reviewers was resolved by consensus or consultation with a third reviewer (YA). In case of missing data or unobtainable articles, the respective author(s) were contacted by email to request further information or the full text if necessary.

## Quality assessment

The risk of bias (RoB) of the included studies was evaluated by three independent reviewers (UAR, NA, and MM). In case of discrepancies or disagreements, a fourth reviewer (YA) was consulted. Due to the lack of a standardised tool to assess the RoB in *in-vitro* studies, a customised tool was developed by adaptation from Raj, Kheur [40] to suit this review. The customised RoB tool comprised seven items as follows: (i) cancer cell lines used; (ii) duration of intervention/ exposure to the cancer cell culture; (iii) concentration used on the cancer cell culture; (iv) culture media used for control; (v) tools used to assess the outcome; (vi) triplication of experiments; and (vii) number of independent experiments performed. Each item was indicated with a "yes" or "no." A study with a score of more than 70% or less than 50% "yes" was judged as low or high risk of bias correspondingly. Any score in between was judged as a moderate risk of bias. Meanwhile, the Systematic Review Centre for Laboratory Animal Experimentation (SYRCLE) tool was used to assess the methodological quality of included animal studies [41]. This RoB tool was adapted from the recommended Cochrane's RoB and has been adjusted to suit aspects of bias that play a specific role in animal intervention studies. The components of the SYRCLE tool assess the following domains: (1) Selection bias: random sequence generation, baseline characteristics, allocation concealment; (2) Detection bias: random housing, blinding, random outcome assessment; (3) Attrition bias: incomplete outcome data; (4) Reporting bias: selective reporting; and (5) Other bias. A study was judged as having a 'low', 'high', or 'unclear' risk of bias accordingly.

## Data synthesis

The baseline characteristics of the included studies were tabulated and described narratively according to the synthesis without meta-analysis (SWiM) reporting guideline [42]. This approach ensures transparency in the event where meta-analysis is not appropriate. Therefore, the included studies were grouped according to the bioactive compound used and were

tabulated according to the primary and secondary outcomes identified. A meta-analysis was not performed due to the lack of suitable data and heterogeneity of the included studies.

## Results

### Study selections

The keyword search performed in three databases resulted in the identification of 1,631 studies, whilst four studies [43–46] were identified through manual screening of the references of eligible studies. After excluding duplicates, 1,283 studies were screened, and 1,088 studies were excluded due to irrelevant titles and/or abstracts. From the remaining 195 studies considered for full-text screening, 160 studies were excluded for the following reasons: 116 studies were dismissed due to wrong publication type (review articles), two studies were excluded due to wrong study design (not *in-vitro/ vivo*), 40 studies were excluded due to wrong intervention (did not report the specific bioactive compounds used, biologically enhanced compound, or combined intervention used were as specified in the inclusion criteria), 1 study was excluded due to wrong comparator (curcumin against its analogue), and 1 study was published in the Chinese language. Twelve studies were excluded as the full text was unobtainable. Attempts were made to obtain the articles, including reaching out to the authors by emailing them. Aside from that, we also requested institutional library services and direct requests to the authors at the Research Gate community. Finally, only 27 studies were included in this review (Fig 1).

### Risk of bias assessment

The risk of bias (RoB) assessment for the *in-vitro* studies included in this review is presented in Fig 2 and S1 Table. The type of cancer cell lines used, duration of intervention/exposure, culture media used for control, and the tools used to assess the outcome were low risk in all the included studies. The domain pertaining to the concentration of cancer cell culture employed was judged to have unclear RoB in 3 of the studies, while the risk in the remaining studies (n = 23) was considered as low. The triplication of experiments was either unclear (n = 11) or low (n = 15) risk of bias. Meanwhile, the number of independent experiments performed was unclear in 3 studies and low risk for the remainder (n = 23). Altogether, only 1 of 26 included *in-vitro* studies were judged as having a moderate risk of bias, while the remaining studies were only low risk.

The SYRCLE risk of bias for reporting *in-vivo* studies is shown in Fig 3 and S2 Table. Selection bias was unclear for all *in-vivo* studies included. Performance and detection bias were high in the included studies. However, attrition and reporting bias were all low in these studies. On the other hand, the other bias was low in one of the studies and unclear for the remainder (n = 2). Altogether, one of the included studies *in-vivo* had a moderate risk of bias, while the remaining (n = 2) were high risk.

### Study characteristics

S3 Table summarises the characteristics of all the included studies. Overall, 27 studies were included in this review. The year of publication of the studies ranged from 2004 to 2023. The majority of included studies were conducted in the Asian region, with eight studies from India [43, 47–53], seven from China [35, 46, 54–58], four from Thailand [45, 59–61], and one from Malaysia [62] and Iran [63] each respectively, whilst the remaining five studies originated from the United States of America (n = 3) [19, 44, 64], Poland (n = 1) [65], Brazil (n = 1) [66], and Mexico (n = 1) [67].

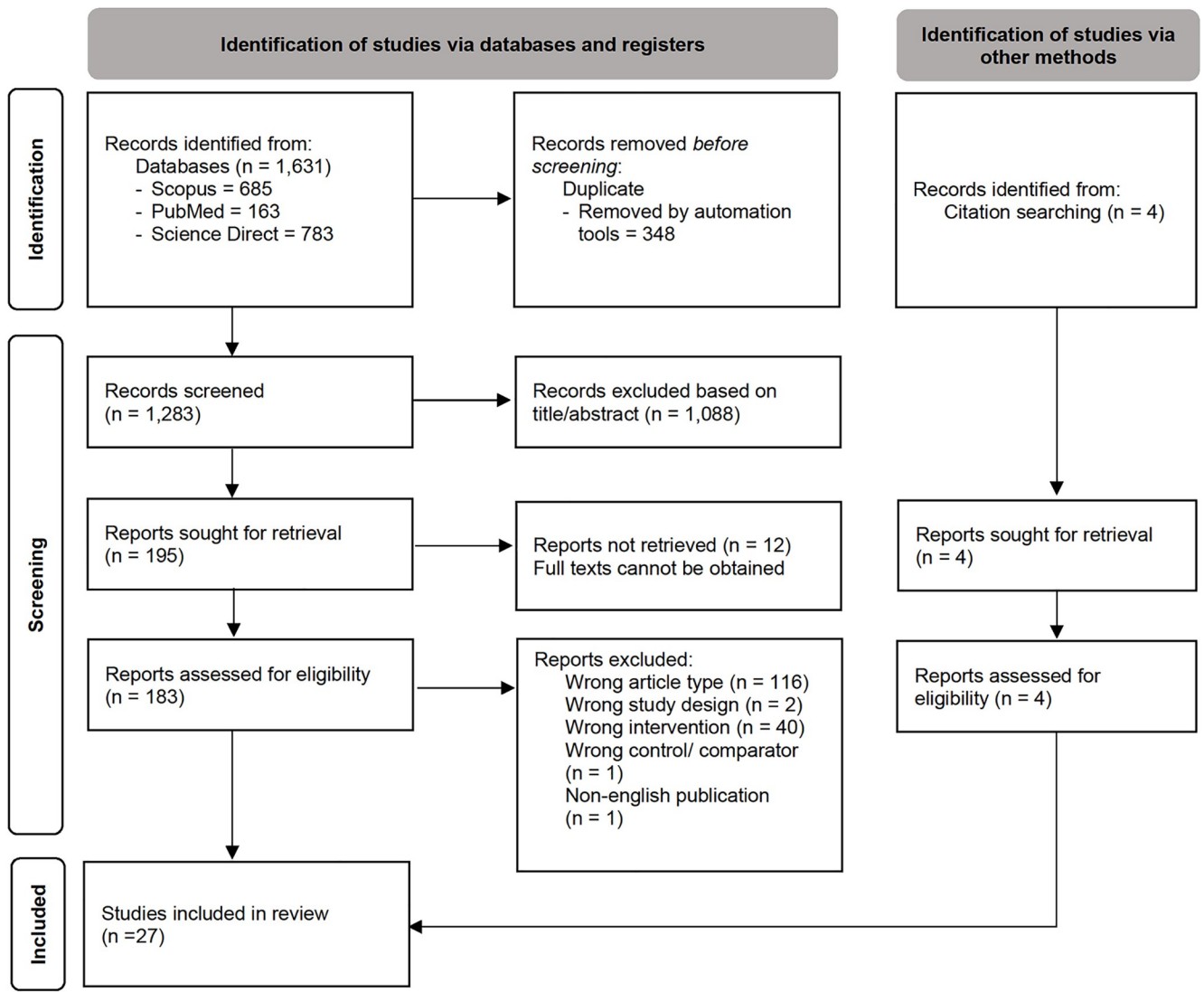

**Fig 1. PRISMA flow diagram depicting article selection process.**

The majority of the included studies were conducted *in-vitro* (n = 24) [1–15, 18, 19, 22–44, 46–50, 52–72], and only one study implemented the *in-vivo* model. The remaining two studies consist of both *in-vitro* and *in-vivo* research models. Various human-derived cervical cancer cell lines were used in the *in-vitro* studies, including HeLa, ME-180, SiHa, C33A, CaSki, SW756, KB-V1, and KB-3-1. The different types of HPV infection in the cells used were reported, such as HPV-18, which was identified in HeLa and SW756 cells, HPV-16 in CaSki and SiHa cells, and HPV-39 in ME-180 cells. The C33A cell line is negative for HPV. Meanwhile, KB-V1 is a multidrug-resistant human cervical carcinoma cell line, whereas KB-3-1 is a drug-sensitive cervical carcinoma cell line.

Risk of bias domain

| Study | D1 | D2 | D3 | D4 | D5 | D6 | D7 | Overall |
|---|---|---|---|---|---|---|---|---|
| Chakrabourty et al., 2012 | + | + | + | + | + | + | + | + |
| Chearwae et al., 2004 | + | + | + | + | + | + | + | + |
| Debata et al., 2013 | + | + | - | + | + | + | + | + |
| Divya et al., 2006 | + | + | + | + | + | + | + | + |
| Ghasemi et al., 2019 | + | + | + | + | + | - | + | + |
| Kapoor et al., 2016 | + | + | + | + | + | + | - | + |
| Lewinska et al., 2014 | + | + | + | + | + | - | + | + |
| Limtrakul et al., 2004 | + | + | - | + | + | + | + | + |
| Liu et al., 2012 | + | + | + | + | + | + | + | + |
| Madden et al., 2009 | + | + | + | + | + | - | + | + |
| Maher et al., 2011 | + | + | + | + | + | + | + | + |
| Martins de Olivera et al., 2023 | + | + | + | + | + | + | + | + |
| Mohammad Noor et al., 2020 | + | + | + | + | + | + | - | + |
| Patiño-Morales et al., 2020 | + | + | + | + | + | - | + | + |
| Pei et al., 2021 | + | + | + | + | + | - | + | + |
| Prusty & Dash, 2005 | + | + | - | + | + | - | - | - |
| Raghav et al., 2018 | + | + | + | + | + | - | + | + |
| Rastogi et al., 2015 | + | + | + | + | + | - | + | + |
| Ruangnoo et al., 2012 | + | + | + | + | + | + | + | + |
| Shang et al., 2016 | + | + | + | + | + | - | + | + |
| Singh et al., 2009 | + | + | + | + | + | - | + | + |
| Singh et al., 2011 | + | + | + | + | + | - | + | + |
| Wang et al., 2020 | + | + | + | + | + | + | + | + |
| Zhang et al., 2017a | + | + | + | + | + | + | + | + |
| Zhang et al., 2017b | + | + | + | + | + | + | + | + |
| Zhao et al., 2023 | + | + | + | + | + | + | + | + |

D1: Cancer cell lines used
D2: Duration of intervention, exposure to the cancer cell culture
D3: Concentration used on the cancer cell culture   4
D4: Concentration used on the cancer cell culture   5
D5: Tools used to assess the outcome
D6: Triplication of experiments
D7: Number of independent experiments performed

Judgement
- Some concerns
+ Low

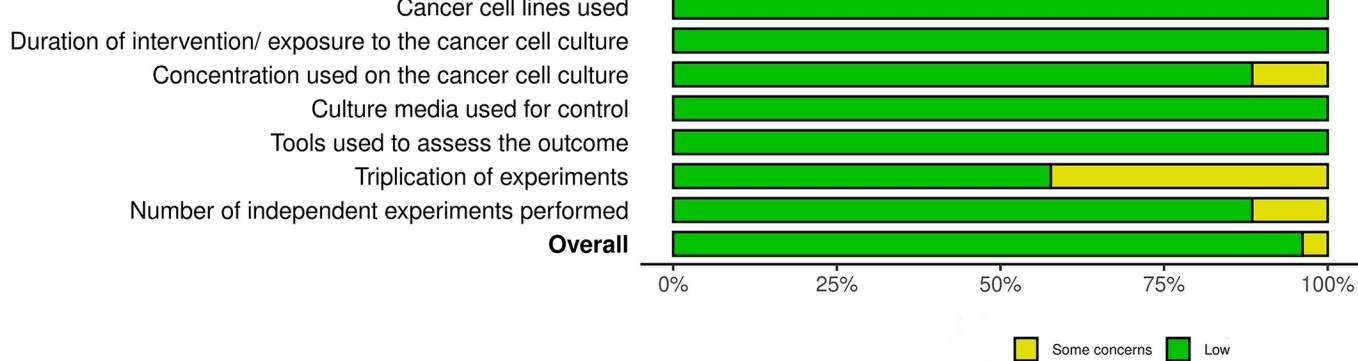

**Fig 2. Customised Risk of Bias (ROB) assessment of included *in-vitro* studies.**

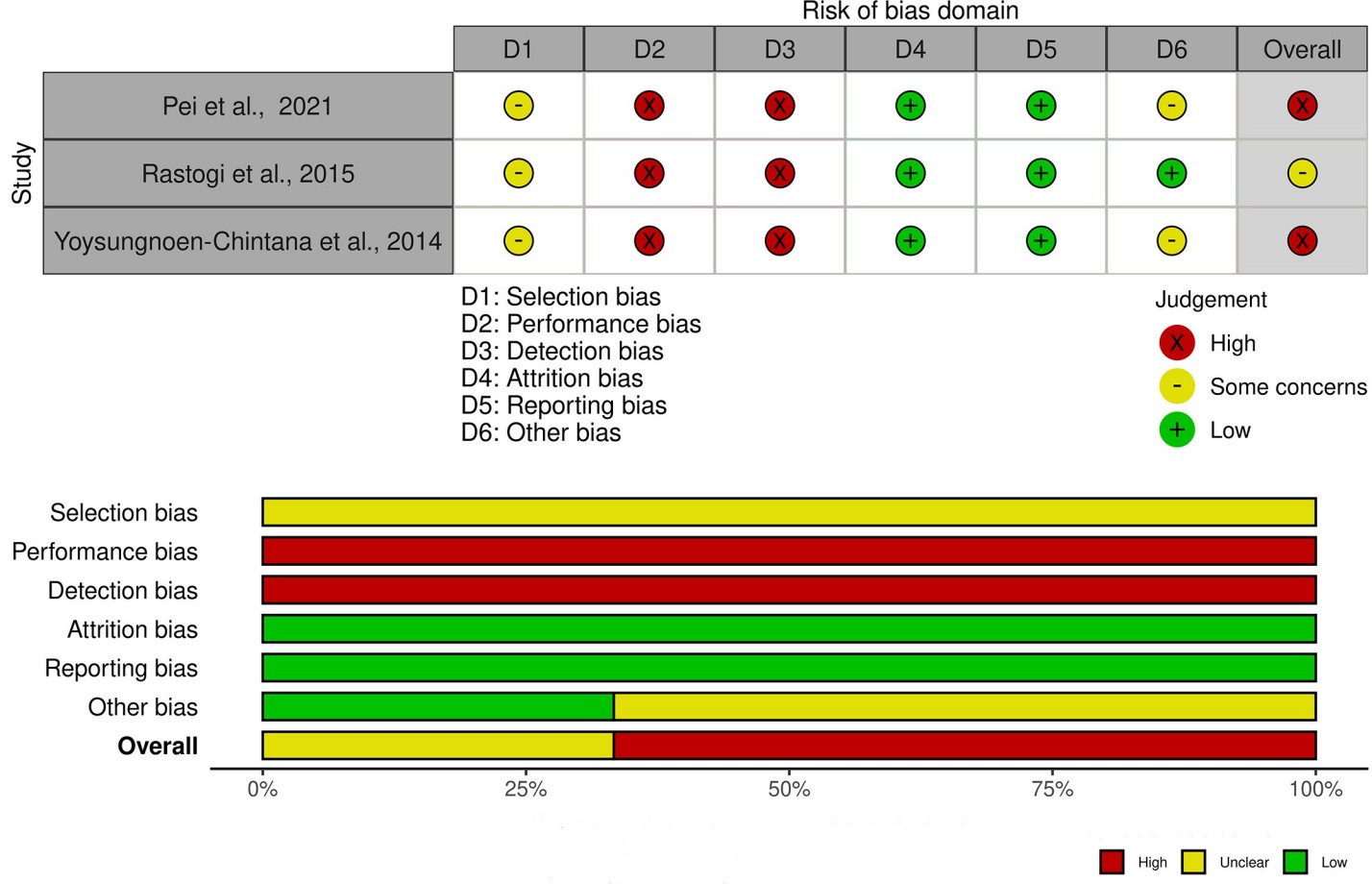

**Fig 3. SYRCLE Risk of Bias (ROB) assessment of included *in-vivo* studies.**

Concerning the type of bioactive compound used in the included studies, the majority of studies investigated the effect of curcumin on cervical cancer (n = 19) [19, 44, 45, 47, 49, 50, 52, 53, 55, 58–60, 62–67, 71]. The remaining eight studies explored the anticancer effects of ginger bioactive compounds, namely 6-gingerol (n = 5) [43, 48, 51, 56, 61], 10-gingerol (n = 1) [56], and 6-shogaol (n = 2) [35, 54]. No studies reported the use of bioactive compounds, as stated above, in combination as a treatment intervention.

For *in-vivo* studies, the type of rodents used in the experiments were BALB/c nude mice (n = 2) and nu/nu nude mice (n = 1). All the mice in the studies were subcutaneously injected with either HeLa or CaSki cell suspension. Treatments were administered through the intra-peritoneal route [51], intragastric [35] or orally [45].

As summarised in Table 1, all studies have demonstrated the potential anti-cancer effects of the bioactive compounds on a variety of cervical cancer cell lines. Different bioactive compounds targeted multiple genes and molecular signalling pathways.

## Discussion

This review comprehensively summarized the anti-cancer potential of curcumin, 6-shogaol, and 6/10-gingerol in pre-clinical studies. The current systematic review has improved upon

**Table 1. Summary of the mechanism of signalling pathways and molecular targets of included studies.** Concentration, dose, and duration of interventions based on the analysis of signalling pathways and molecular targets in cervical cancer cells.

| Compound | Cell type | Intervention | | Mechanism of signalling & molecular targets | References |
|---|---|---|---|---|---|
| | | Concentration | Duration | | |
| Curcumin | KB-V1 | 0–30 μM | 72 hr | ↓ MDR-I gene expression<br>↓ Pgp<br>↑ Pgp substrate; calcein-AM, rhodamine123, and bodipy-FL-vinblastine<br>↑ ATPase activity | [59, 60] |
| | HeLa | 0.1–5 μM | 24 hr | ↑ superoxide levels via global DNA hypermethylation<br>↑ ROS production by targeting the cytosolic/nuclear thioredoxin system leading to apoptosis | [65] |
| | | 0–15 μM | 24–48 hr | ↓ Ezrin (VIL2) protein<br>↓ Phosphoglycerate kinase (PGK1)<br>↑ stress-induced phosphoprotein (STIP)<br>↑ Rad50 protein<br>↑ enolase isoforms 7402 (ENO1 7420 and ENO1 7407)<br>↑ UROD and FSIP2 protein expression<br>DNA damage response | [64] |
| | | 5–20 μM | 24 hr | ↓ E6-associated protein (E6AP)-p53 interaction<br>↓ E6 and E7 oncogene expression<br>↓ COX-2<br>↓ TNF-α induced NF-κB activation<br>↑ NQ01<br>↑ p21 and p53<br>Activation of Keap1/Nrf2 pathway<br>Cytotoxicity via NQO1-p53 complex formation<br>Prevents AP-1 binding<br>Inactivation of NF-κB pathway | [47, 67] |
| | | 10–15 μM | 18–20 hr | ↓ ATPase activity of mitotic kinesin Eg5<br>Inhibits centrosomal separation<br>Induces mitotic arrest | [50] |
| | | 13 μM | 0–48 hr | ↑ p-ATM and p-ATR<br>↑ p53<br>↑ MDM2<br>↑ BRCA1<br>↑ DNA-PK<br>↑ MDC1<br>↑ p-H2A.X<br>↑ PARP<br>↑ MGMT protein expression<br>DNA damage response | [55] |
| | | 20–50 μM | 24 hr | ↓ p16<br>↓ oncogene E6 and E7<br>↓ PCNA<br>↓ cyclin D1<br>↓ caspase-3, -9<br>↓ Becn1<br>↓ PARP1<br>↓ Bcl-2<br>↓ N-cadherin, E-cadherin, vimentin<br>↑ p53, p21 and p73<br>↑ Bax | [53, 58, 66] |
| | | 34.23 μM | 48 hr | ↓ NF-κB<br>↓ Wnt/b-catenin | [63] |
| | | 50 μM | 8 hr | ↓ E6 oncogene<br>↓ EGFR<br>↓ Rb phosphorylation activation at serine-780 phosphorylation<br>↑ p53 | [19] |
| | | 50–100 μM | 24 hr | ↓ ERK and Ras<br>↓ cell cycle genes cyclin (D1 and Hsp 70)<br>↓ inflammatory proteins (Cox-2 and iNOS)<br>↓ p53<br>↓ Bcl-2 and Bcl-XL<br>↓ C-Myc<br>↑ JNK<br>↑ Bax<br>↑ cytochrome c and caspase-3, -9<br>Targets MAPK/ERK pathway | [52] |
| | | 100 μM | 15 mins- 6 hr | ↓ p53-responsive gene<br>↓ WAF-1/p21 expression<br>Interfere with AP-1 binding activity via ↓ c-fos and ↑ fra-1 expression | [49] |

*(Continued)*

**Table 1.** (Continued)

| | | | | |
|---|---|---|---|---|
| **SiHa** | 5–20 µM | 24 hr | ↓ E6 and E7 oncogene expression<br>↓ TNF-α induced NF-κB activation<br>↓ E6-associated protein (E6AP)-p53 interaction<br>↑ NQ01<br>↑ p21 and p53<br>Activation of Keap1/Nrf2 pathway<br>Inactivation of NF-κB pathway<br>Cytotoxicity via NQO1-p53 complex formation | [47, 67] |
| | 20–50 µM | 24–48 hr | ↓ oncogene E6 and E7 mRNA<br>↓ PCNA<br>↓ cyclin D1<br>↓ G2/M-related genes (cyclins B1 and cdc25)<br>↑ cleaved caspase-3 and PARP protein<br>↑ p62<br>↑ LC3I and LC3II leading to autophagy<br>↑ p53 and p21 | [44, 46, 53] |
| | 50 µM | 8 hr | ↓ oncogene E6<br>↓ EGFR<br>↑ p53 | [19] |
| | 50–100 µM | 24 hr | ↓ ERK<br>↓ cyclin D1 and Hsp 70<br>↓ p53 and p73<br>↓ Cox-2 and iNOS<br>↓ Bcl-2 and Bcl-XL<br>↓ C-Myc<br>↑ JNK<br>↑ Bax<br>↑ cytochrome c and caspase-3, -9<br>Targets MAPK/ERK pathway | [52] |
| **ME-180** | 50 µM | 8 hr | ↓ E6 oncogene<br>↓ EGFR<br>↓ Rb phosphorylation activation at serine-780 phosphorylation<br>↑ p53 | [19] |
| **SW756** | 50 µM | 8 hr | ↓ E6 oncogene<br>↓ EGFR<br>↑ p53 | [19] |
| **C33A** | 5–50 µM | 24 hr | ↓ E6 and E7 oncogene expression<br>↓ TNF-α induced NF-κB activation<br>↓ p16<br>↓ cyclin D1<br>↑ p53 and p73 gene expression<br>Inactivation of NF-κB pathway | [47, 53] |
| **CaSki** | 10–50 µM | 24 hr | ↓ E6-associated protein (E6AP)-p53 interaction<br>↓ PCNA<br>↓ cyclin D1<br>↓ oncogene E6 and E7<br>↑ NQ01<br>↑ p21 and p53<br>Cytotoxicity via NQO1-p53 complex formation<br>Activation of Keap1/Nrf2 pathway | [53, 67] |
| | 20–40 µM | 24–48 hr | ↓ oncogene E6 and E7<br>↓ BaP-induced HPV E7<br>↑ caspase-3, -9 and cleaved PARP<br>↑ p53, pRb, and PTPN13 expression | [44] |
| | 50–100 µM | 24 hr | ↓ ERK<br>↓ cyclin D1 and Hsp 70<br>↓ p53 and p73<br>↓ C-Myc<br>↑ JNK<br>↑ Bax, cytochrome c and caspase-3, -9<br>Targets MAPK/ERK pathway | [52] |

(*Continued*)

**Table 1.**  (Continued)

| Compound | Cell type | Dosage | Duration | Mechanism of signalling & molecular targets | References |
|---|---|---|---|---|---|
| 6-gingerol | HeLa | 50 µM | 24 hr | ↓ cyclin B1<br>↓ of PARP<br>↑ p53 and p21<br>↑ p-H2AX<br>↑ cleaved caspase-3<br>Genotoxic stress via MRC-1 inhibition<br>DNA damage response | [51] |
| | | 60–140 µM | 48 hr | ↓ cell cycle genes (cyclin A, cyclin D1, cyclin E1) and CDK-1<br>↓ p21 and p27<br>↓ PI3K/AKT pathway<br>↑ cytochrome c, Bax, caspase-3, -8, -9, and PARP<br>↑ of AMPK | [57] |
| | | 75–125 µg/mL | 48 hr | ↓ NF-κB<br>↓ AKT<br>↑ caspase 3, PARP, Bax and cytochrome c<br>↑ Bcl-2<br>↑ TNFα<br>Induce autophagy death via ↑ acidic vacuoles (AVO) formation | [43] |
| | CaSki | 50 µM | 24 hr | ↓ cyclin B1<br>↓ PARP expressions<br>↑ p53 and p21<br>↑ p-H2AX<br>↑ cleaved caspase-3<br>Genotoxic stress via MRC-1 inhibition<br>Oncogene E6 and E7 shows no changes | [51] |
| 10-gingerol | HeLa | 15–50 µM | 48 hr | ↓ cyclin A, cyclin D1, cyclin E1, CDK-1, CDK-2, CDK-4, CDK-6<br>↓ p15, p21, p16, and p27<br>↓ GSK-3B<br>↓ β-catenin<br>↓ Bcl-2<br>↓ PI3K/AKT pathway<br>↓ mTOR phosphorylation<br>↓ NF-κB pathway<br>↑ death receptors proteins DR3 and DR5<br>↑ cleaved caspase-3, -8, -9, cytochrome c, Bid, Bad, and Bax<br>↑ AMPK pathway | [56] |
| 6-shogaol | HeLa | 20–40 µM | 0–24 hr | ↓ CDC25A<br>↓ cyclin B1<br>↓ PCNA<br>↓ Bcl-2<br>↓ p62<br>↓ cell migration regulators genes (N-cadherin, MMP-2, and MMP9)<br>↓ Snail, Twist, Zeb-1, and Zeb-2 gene expression<br>↓ PI3K/AKT/mTOR pathway via ↓ p-PI3K, p-AKT, and p-mTOR<br>↓ ER-stress-associated protein (PERK and ARF5)<br>↑ cytochrome c, PARP, Bax, and caspase 3<br>↑ autophagy-related proteins (LC3-II and Beclin 1)<br>↑ E-cadherin<br>↑ HSP60<br>↑ Annexin A1, cofilin, and calreticulin<br>No changes in CHOP expression | [35, 54] |
| | SiHa | 20–40 µM | 24 hr | ↓ Bcl-2<br>↓ p62<br>↓ cell migration regulators genes (N-cadherin, MMP-2, and MMP9)<br>↓ Snail, Twist, Zeb-1, and Zeb-2 gene expression<br>↓ PI3K/AKT/mTOR pathway via ↓ p-PI3K, p-AKT, and p-mTOR<br>↑ cytochrome c, PARP, and Bax<br>↑ autophagy-related proteins (LC3-II and Beclin 1)<br>↑ E-cadherin | [35] |

| In-vivo studies | | | | | |
|---|---|---|---|---|---|
| Compound | Cell type | Intervention | | Mechanism of signalling & molecular targets | References |
| | | Dosage | Duration | | |
| Curcumin | CaSki implanted nude mice | 1,500 mg/Kg | 0–28 days | ↓ angiogenesis biomarkers (VEGF, COX-2, EGFR) | [45] |

(*Continued*)

**Table 1.** (Continued)

| 6-gingerol | HeLa implanted mice | 2.5-5mg/Kg | 0–30 days | ↓ Bax, GADD45, Noxa, Puma<br>↓ p21<br>↑ oxidative stress biomarker (MDA)<br>↑ p53 | [51] |
| --- | --- | --- | --- | --- | --- |

↑: upregulation, ↓: downregulation, MDR-1: multidrug resistance 1, Pgp: phosphoglycolate phosphatase, ATPase: adenosine triphosphatase, ROS: reactive oxygen species, STIP: stress-induced phosphoprotein, PGK1: Phosphoglycerate kinase-1, ENO1: enolase 1, UROD: uroporphyrinogen decarboxylase, FSIP2: fibrous sheath interacting protein, Keap1: Kelch-like ECH-associated protein 1, Nrf2: nuclear factor erythroid 2-related factor 2, NQ01: NAD(P)H quinone dehydrogenase 1, E6AP: E6-associated protein, COX-2: cyclooxygenase-2, AP-1: activator protein-1, TNF-α: tumour necrosis factor alpha, NF-κB: nuclear factor kappa B, ATM: ataxia telangiectasia mutated, ATR: ataxia telangiectasia and Rad3-related protein, MDM2: murine double minute 2, BRCA1: breast cancer gene 1, DNA-PK: DNA-dependent protein kinase, MDC1: mediator of DNA damage checkpoint protein 1, p-H2A.X: phosphorylated histone H2A Variant X, PARP: poly(ADP-ribose) polymerase, MGMT: O6-methylguanine-DNA methyltransferase, PCNA: proliferating cell nuclear antigen, Becn1: Beclin 1, Bcl-2: B-cell lymphoma 2, Bax, Bcl-2 associated X protein, N-cadherin: Neural cadherin, E-cadherin; Epithelial cadherin, Wnt: wingless and Integrated-1, EGFR: epidermal growth factor receptor, Rb: retinoblastoma, MAPK: Mitogen-activated protein Kinase, ERK: Extracellular signal-regulated kinase, Ras: Rat sarcoma viral oncogene, JNK: c-Jun N-terminal kinase, Hsp70: Heat shock protein 70, iNOS: inducible nitric oxide synthase, Bcl-xl: B-cell lymphoma extra-large, C-myc: myelocytomatosis oncogene, c-fos: cellular proto-oncogene fos, fra-1: Fos-related antigen 1, WAF-1: wild-type p53-activated fragment, MRC-1: macrophage mannose receptor-1, CDK-1: cyclin-dependent kinase-1, AMPK: adenosine monophosphate-activated protein, AKT: protein kinase B, GSK-3B: glycogen synthase-3 beta, PI3K: phosphoinositide 3-kinase, mTOR: mechanistic target of rapamycin, CDC25A: cell division cycle 25A, MMP: matrix metalloproteinase, Zeb: Zinc finger E-box-binding homeobox, HSP60: heat shock protein 60, CHOP: C/EBP homologous protein, VEGF: Vascular endothelial growth factor, GADD45: growth arrest and DNA damage-inducible 45, MDA: malondialdehyde

previous reviews as it refined its focus by including only studies using bioactive compounds, thus ensuring uniformity of intervention. Previous reviews concentrated on specific extracts like ginger and turmeric or for addressing the bioactive compounds in different cancer types or other health conditions(s), many of which lacked systematic methodologies. The collated evidence has shown that curcumin, 6/10-gingerol, and 6-shogaol exerted their anti-cancer activity via the disruption of several molecular pathways and by targeting certain genes and protein expression.

## The anti-cancer activities and cytotoxicity effects of bioactive compounds

**Cancer cell proliferation and viability.** The cytotoxic effect of curcumin, 6/10-gingerol and 6-shogaol carried out in different cervical cancer cell lines is shown to reduce the cell viability of cancer cells in a concentration- and time-dependent manner. The anti-proliferative activities across the different bioactive compounds showed varied results on cervical cancer cells. Many studies have reported that the highest concentration of these compounds is associated with the lowest cell proliferation in all tested cell lines [35, 43, 44, 47, 48, 50, 51, 54, 65, 66]. This observation indicates that cell proliferation inhibition is concentration-dependent. Hence, the $IC_{50}$ value (half maximal inhibitory concentration) is used to inform the researchers on the amount of drug or bioactive compounds required to inhibit cell proliferation by half.

Evidence has shown that treatment with curcumin caused a significant inhibition in cervical cancer cell proliferation, including HPV-infected (i.e., HPV-16, HPV-18, and HPV-39) and HPV-negative cervical carcinoma within 24 to 96 hr [19, 44, 47, 55, 58, 59, 62–67, 71]. The $IC_{50}$ values ranged between 7 to 34 μM in HeLa cells. Similarly, cell viability reduction in HeLa cell lines was observed after treatment with 6-gingerol [43, 48, 51, 57, 61], 10-gingerol [56] and 6-shogaol [31, 50] at varying $IC_{50}$ values ranging from 0.6 to 431 μM within 24 to 96 hr. Based on the review findings, it is difficult to gauge the optimal $IC_{50}$ to inhibit cervical cells proliferation due to several factors including the type of cell lines used, the duration of exposure to the studied bioactive compound(s) and the culture media used. The cell viability assay used may

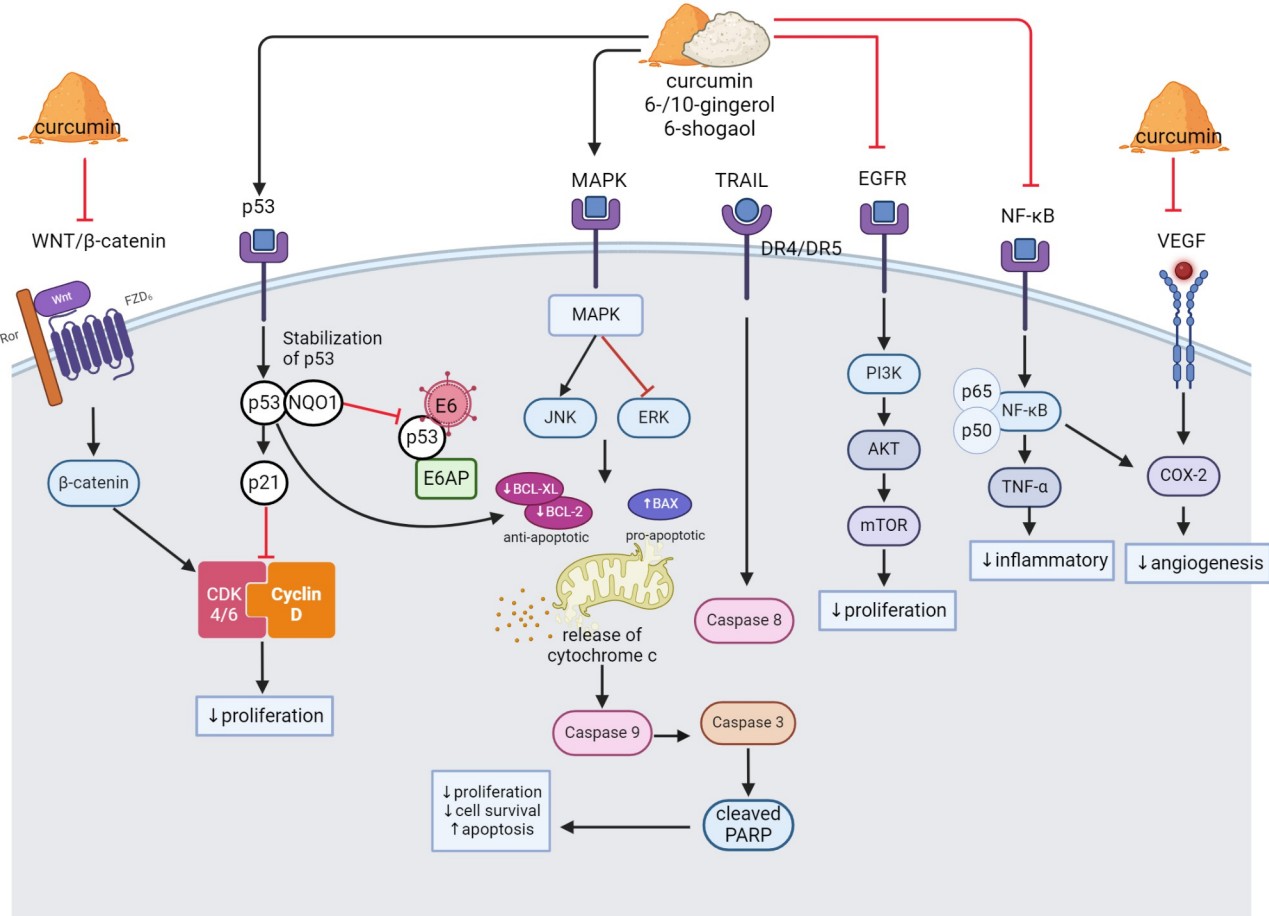

**Fig 4. The anti-cancer signalling pathways modulated by curcumin, gingerol, and shogaol in cervical cancer cells.** (1) Curcumin inhibits cell proliferation via inhibition of the WNT/β-catenin pathway. β-catenin causes overexpression of CDK 4/6 that are involved in the cell cycle. (2) Curcumin, gingerol, and shogaol induce the p53/p21 signalling pathway by promoting stability and restoration of p53 tumour suppressor activity, therefore decreasing cell proliferation. The restoration of p53 inhibits oncogene E6 from ubiquitination of the p53 protein (3) Curcumin, gingerol, and shogaol target MAPK/ERK pathways by upregulation of JNK and downregulation of ERK proteins leading to apoptosis. (4) Curcumin, gingerol, and shogaol promote intrinsic apoptotic pathways by upregulation of pro-apoptotic protein expression, including BAX, caspase-3, caspase-9, and cleaved PARP, while downregulates anti-apoptotic protein BCL2 and BCL-XL. 10-gingerol promotes extrinsic apoptotic pathway by modulating death receptor protein DR4/DR5. (5) Curcumin, gingerol, and shogaol suppress the PI3K/AKT/mTOR pathway. (6) Curcumin, gingerol, and shogaol reduced cell inflammation by impairing the NF-kB signalling pathway, leading to downregulation of COX-2.

also influence the measurements on cervical cancer cell lines based on the cell seeding number, assay concentration and incubation time, serum starvation, released intracellular contents, and extrusion of formazan to the extracellular space [73].

Inflammation and NF-κB have been found to be closely related to cancer cell proliferation which promotes cell proliferation and survival, as well as tumour growth [69]. Curcumin was reported to downregulate the expression of tumour necrosis factor-alpha (TNF-α) and impaired the NF-κB signalling pathways, leading to the downregulation of cyclooxygenase-2 (COX-2) expression in HeLa, SiHa and C33A cell lines [47]. These findings were supported by other studies that demonstrated the inhibition of inflammatory NF-κB signalling pathway in HeLa, ME-180, SiHa, C33A and SW756 cell lines [19, 44] and HeLa 3-D culture models [63, 66]. In addition to its effects on the NF-kB pathway, curcumin has been reported by Ghasemi et al. [63] to target and impair Wnt/β-catenin (WNT) signalling. Meanwhile, Singh and Singh [52] reported that curcumin targets MAPK/ERK pathways, which are involved in cell growth

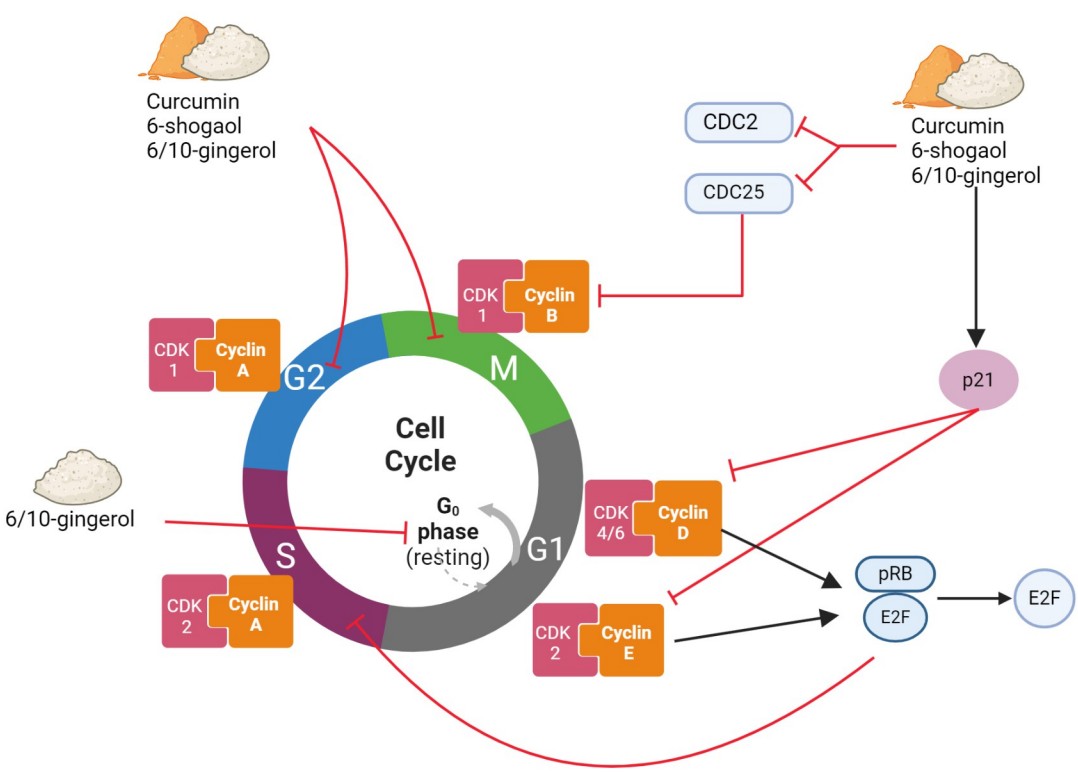

**Fig 5. Cell cycle arrest induced by curcumin, gingerol, and shogaol in cervical cancer cells.** (1) Curcumin, gingerols, and shogaol induced cell cycle arrest at the $G_2$/M phase. (2) Curcumin, 6-shogaol and gingerol downregulate the expression of CDC25 and CDC2 proteins, while upregulate p21. The downstream effect of p21 inhibits CDK2 and CDK4, preventing the phosphorylation of pRb, which will bind to E2F transcription factors and prohibit cells from entering the S phase. (3) Gingerol-induced cell cycle arrest at the $G_0$/$G_1$ phase by downregulating cyclin A, D, and E and CDK gene expression.

and survival in HeLa, SiHa and CaSki cell lines. Furthermore, curcumin promotes stability and restoration of p53 protein via interaction with NAD(P) H: quionone oxidoreductase 1 (NQO1) protein, increases the p53 half-life, and therefore decreases cell viability of cervical cancer cells [67]. The binding of p53-NQ01 avoids the interaction between p53 and its negative regulator ubiquitin ligase E6-associated protein, subsequently activate the p53 apoptotic pathway in cervical cancer cell lines [67]. The possible anti-cancer signalling pathways by curcumin, gingerol and shogaol is illustrated in Fig 4.

A study by Chakraborty et al. [43] reported that 6-gingerol downregulated the expression of NF-kB as well as AKT protein kinase. This finding was supported by Zhang et al. [57] that demonstrated inhibition of the PI3K/AKT pathway involved in inducing mTOR-mediated cell apoptosis and activation of AMPK phosphorylation. Additionally, a similar effect was observed from treatment with 10-gingerol on HeLa cell lines in a study reported by Zhang et al. [56]. Meanwhile, Pei et al. [35] demonstrated that 6-shogaol targets and downregulates protein expression of p-PI3K, p-AKT, and p-mTOR overall, consequently suppressing the PI3K/AKT/mTOR pathway leading to a reduction in cell proliferation and viability in HeLa and SiHa cells.

**Cancer cell apoptosis.** Apoptosis is a programmed cell death, and findings from this review elucidate the role of curcumin, 6/10-gingerol and 6-shogaol in triggering apoptosis through various mechanisms. One of the common mechanisms involves the upregulation of the caspase family, including caspase-3, caspase-8, and caspase-9, which play a role in the apoptotic process [44, 46, 47, 52, 55, 63, 65]. These proteins are crucial in the activation of the downstream apoptosis process and cleaving of poly ADP-ribose polymerase (PARP), an apoptotic substrate, therefore suggesting the role of mitochondria in apoptosis [44]. Reduction of mitochondrial membrane potential (MMP) caused by curcumin and 6-gingerol, as demonstrated by Zhao et al. [58] and Chakraborty et al. [43], suggests the critical role of mitochondrial pathways in their apoptosis-inducing effects. In addition to caspase upregulation, curcumin, 6/10-gingerol, and 6-shogaol also has been found to downregulate the expression of anti-apoptotic protein including Bcl-2 and Bcl-XL while upregulating the pro-apoptotic Bax protein [43, 48, 51, 56–58]. The apoptotic signalling pathways induced by curcumin, gingerol and shogaol are illustrated in Fig 4.

Furthermore, Rastogi et al. [51] has reported that 6-gingerol upregulated tumour suppressor protein, p53 and p21, via inhibition of the proteasome activity and induced oxidative stress in the cervical cells. On the other hand, 10-gingerol has been shown to modulate death receptor proteins, DR3 and DR5, which are involved in the extrinsic apoptotic pathway [56].

It is also known that an increase in JNK and ERK protein expression plays an important part in apoptosis [68]. Interestingly, curcumin is reported to downregulate ERK expression but upregulate JNK expression [52]. The upregulation of JNK by curcumin may lead to the activation of downstream apoptotic signalling pathways, including Bax and AIF (apoptosis-inducing factor), and the release of cytochrome c from the mitochondria. Whereas the downregulation of ERK by curcumin may inhibit cell survival and consequently promote apoptosis [52]. Based on the reported findings presented, there is evidence that curcumin, 6/10-gingerol, and 6-shogaol are capable of inducing apoptosis via extrinsic or intrinsic apoptotic pathways, particularly the caspase-dependent mechanism. Nevertheless, there is a lack of evidence as to whether these bioactive compounds induce caspase-independent apoptosis. The caspase-independent apoptosis pathway may be regulated by a family of serine proteases known as granzymes. Granzyme A elicits apoptosis independent of the caspases, whereas granzyme B may trigger apoptosis via caspase-dependent and -independent mechanisms. Another caspase-independent pathway involves a family of cysteine proteases, namely the calpains (calcium-activated neutral proteases) [74].

**Cancer cell cycle arrest.** Curcumin, 6/10-gingerol and 6-shogaol have been shown to modulate the cancer cell cycle, resulting in the inhibition of cell proliferation. Curcumin [63, 65, 71], 6-, 10- gingerols [48, 51], and 6-shogaol [35, 54] induced cell cycle arrest at the $G_2$/M phase in SiHa and HeLa cell lines. This cell cycle arrest is associated with the downregulation of cell cycle genes, cyclin D1 and cyclin B1, leading to inhibition of the cell proliferation [46, 52, 53]. Cyclin D1 is a proto-oncogene involved in cell cycle progression from the G1 to the S phase whereas cyclin B1 forms a complex with cyclin-dependent kinase 1 (CDK1) which is required for the progression of cells into mitosis [70]. The possible cell cycle arrest is illustrated in Fig 5.

Additionally, curcumin and 6-shogaol also downregulate phosphoglycerate kinase (PGK1), a phosphate-transferring enzyme involved in cell growth and decrease the expression of CDC25 and CDC2 proteins while increasing the expression of p21 [46, 64]. The downstream effect of p21 inhibits CDK2 and CDK4, preventing them from phosphorylating the retinoblastoma protein (pRb). As a result, pRb binds to E2F transcription factors and prohibits cells from entering the S phase of the cell cycle, which leads to irreversible cell cycle arrest [46]. In

addition to that, 6-shogaol downregulates proliferating cell nuclear antigen (PCNA) protein expression levels, which are involved in DNA replication and cell proliferation [35].

Furthermore, studies by Zhang et al. [56] and Zhang et al. [57] demonstrated that 6-gingerol and 10-gingerol induced cell cycle arrest at the G0/G1 phase in the HeLa cell line. It was shown that both compounds caused cell cycle arrest by downregulating cell cycle gene expression namely cyclin A, cyclin D1, and cyclin E1 [56].

**Multidrug resistance and drug sensitivity.**   Multidrug resistance (MDR) is where the cancer cells become resistant to multiple drugs, making treatments more challenging. Curcumin has been reported to be an effective MDR regulator in two studies by Chearwae et al. [59] and Limtrakul et al. [60]. The mechanisms responsible for multidrug resistance (MDR) have been linked to the expression of the MDR1 gene product, P-glycoprotein (Pgp). Curcumin has been reported to block the function of Pgp that transports anti-cancer drug substrate in drug-resistant cell lines, KB-V1 [60].

This review also showed that curcumin and 6-gingerol have the potential to enhance the effectiveness of chemotherapy drugs in cervical cancer treatment. A study by Chearwae et al. [59] reported that curcumin increase the sensitivity of KB-V1 cells towards vinblastine, hence suggesting a potential synergistic effect between the two compounds in the treatment of cervical cancer. Meanwhile, Kapoor et al. [48] and Rastogi et al. [51] have observed that 6-gingerol potentiates the cytotoxicity effect of cisplatin in HeLa cells [48, 51] where it induced a maximum apoptosis rate of 72.3%.

**Cancer angiogenesis and metastasis.**   A study by Maher et al. [44] reported that curcumin decreases the cancer cell motility as it rescues protein tyrosine phosphatase non-receptor type 13 (PTPN13), in which the expression of PTPN13 is associated with decrease in anchorage-independent growth. Loss of PTPN13 expression also has been associated with increased invasiveness in cancers [72]. Meanwhile, Pei et al. [35] has reported that 6-shogaol affects the expression of cell migration key regulators, N-cadherin, MMP-2, and MMP-9, thus suppressing the cancer cell migration in HeLa and SiHa cell lines.

**Tumour growth.**   Several studies have demonstrated the effects of curcumin and ginger bioactive compounds on tumour growth involving different mechanisms and pathways. An *in-vivo* study using CaSki-implanted BALB/c-nude mice reported a reduction in tumour volume after receiving curcumin as a treatment [45]. The postulated mechanism involves the anti-angiogenesis effects of curcumin via the downregulation of vascular endothelial growth factor (VEGF), an angiogenesis biomarker, and cyclooxygenase-2 (COX-2) in the EGFR signalling pathways. In a different animal model using 6-gingerol, reduction in tumour volume and tumour height in HeLa-implanted (nu/nu) nude mice was observed via reactivation of p53 and oxidative stress biomarker malondialdehyde (MDA) through proteasomal inhibition [51]. Pei et al. [35] has subjected HeLa xenograft BALB/c nude mice to 6-shogaol. The study reported that 6-shogaol inhibited tumour growth and induced apoptosis without causing body weight loss and/or damage to important organs.

## Study limitations

Nevertheless, this systematic review is not without limitations. Most of the included studies employed an *in-vitro* model. *In-vitro* cell culture may not account for the interactions between physiological processes, metabolic processes, and other cellular mechanisms that occur in an organ system when subjected to the studied bioactive compounds. Consequently, the limitations of *in-vitro* studies may impact the generalizability of findings to clinical settings. The limited robustness of preclinical research has been identified as one of the drivers of clinical trial failure [75]. Without validation through *in-vivo* studies, findings from *in-vitro* may lead to

overestimations or misinterpretations of the efficacy and safety of the treatments when applied to human subjects. Although the overall risk of bias for most in-vitro studies assessed was judged as low, indicating reliable study outcomes, additional *in-vivo* studies are necessary to ascertain the effectiveness and safety profile of the bioactive compounds. Furthermore, the exclusion of 12 studies due to unavailability may have hindered a comprehensive evaluation of the included studies and skewed the overall assessment of their chemotherapeutic potentials on the anti-cancer effects of the compound of interest. Another limitation is a lack of research on the combination of curcumin, gingerol and/or shogaol in the treatment of cervical cancer. Combination treatment has been shown to enhance cytotoxicity towards cervical cancer cells and reduce the required concentrations of compounds/drugs, leading to fewer side effects.

## Future research directions

In this review, the reporting of $IC_{50}$ values reveal significant variability, indicating inconsistencies of $IC_{50}$, which could result in irreproducible experimental outcomes. The systematic $IC_{50}$ errors caused by uneven cell proliferation were found to be the most significant in terms of their misleading effects [76]. It was reported that the $IC_{50}$ values for cisplatin could vary significantly based on cell density and proliferation potential, highlighting the importance of controlling this variable. It was proposed by He and his team that inconsistencies in $IC_{50}$ values are natural properties of cancer cells rather than remedial artifacts [76]. Nevertheless, there should be a concerted effort to improve the reporting of $IC_{50}$, especially in pharmacotherapeutic studies, to ensure consistency and reliability in research findings. Previously, Haibe-Kains et al. [77] suggested the establishment of an international standard for definite IC50 in each cancer cell line, whereas, Punyamurtala et al. [78] described density-dependent IC50 variations as a hallmark of cancer cells and proposed that IC50-seeding density slope (ISDS) serve as a standardized method for assessing therapeutic treatment.

Challenges may exist related to the isolation and purification of these bioactive compounds due to the complexity of the extraction process involved. Among the 27 studies reviewed, 17 reported using commercially available bioactive compounds (purity $\geq$95%). Meanwhile, the remaining studies reported self-isolation and purification, with 4 out of 10 studies reporting a purity of $\geq$90%, indicating a high level of consistency in the quality of compounds used. The focus on the bioactive compounds allows for a more consistent evaluation of efficacy across studies if the purity levels are maintained. Therefore, while challenges related to the isolation and purification of these compounds may exist, the focus on the bioactive compounds rather than varying compositions of the extraction yield could minimise the challenges related to the reproducibility and scalability of research findings.

Despite the evidence presented in this review, there are still potential areas to be explored in term of understanding the anticancer effects of the studied bioactive compounds. For example, future studies should consider exploring the long-term anticancer effect(s) of curcumin, gingerol(s), and/or shogaol(s). Do they cause withdrawal effects on the cervical cancer cells' viability, and will this affect the cancer cells' sensitivity towards conventional chemotherapy? Most of the time, cancer therapy involves long-term treatment. Thus, a safe yet effective treatment is warranted. Moreover, the lack of studies investigating the synergistic effects of curcumin, gingerols, and shogaols when used in combination justifies further research. Given their diverse mechanisms of action, it would be imperative to investigate any potential of synergy in the combination treatments and explore their potential for use in other cancer types. Additionally, mutations in certain mechanisms, such as apoptosis, may lead the cervical cancer cells to develop chemoresistance. Identifying and understanding the molecular mechanism and the targeted protein involved in chemoresistant is crucial for developing effective therapeutic

cervical cancer. By targeting specific pathways has shown potential to enhance chemosensitivity in cervical cancer. Therefore, it is essential to explore other proteins and pathways that may contribute to overcome the resistance mechanisms. This could help with the development of targeted therapies aimed at improving the efficacy of bioactive compounds against chemoresistant cervical cancer cells. Furthermore, future research should be directed to investigate the potential of gingerol and shogaol in modulating MDR cancer cell lines. Future research should focus on investigating the potential of ginger and turmeric bioactive compounds as modulators of multidrug resistance while simultaneously exploring other genes that may contribute to overcoming these resistance mechanisms. The susceptibility of chemoresistant cervical cancer cells towards these natural bioactive compounds may be explored to identify their potential as an adjunct or alternative cancer chemotherapy. Ultimately, the shift from preclinical studies to clinical studies is necessary to translate the anticancer effects of curcumin, gingerols, and shogaols in humans with cervical cancer.

## Conclusions

This review suggests that curcumin, 6-, 10-gingerols and 6-shogaols have promising prospect to be further explored and developed into therapeutic agents for cervical cancer. Evidence have shown that these bioactive compounds possess antiproliferative effects, induce apoptosis and cell cycle arrest, inhibit angiogenesis and metastasis as well as increase the sensitivity of cancer cells to drugs. Nevertheless, further studies are warranted to explore the synergistic effects of these bioactive compounds, especially in *in-vivo* applications, to determine their therapeutic potential on cervical cancer.

## Supporting information

**S1 Table. Risk of bias assessment of *in-vitro* studies.**
(DOCX)

**S2 Table. Risk of bias assessment of *in-vivo* studies.**
(DOCX)

**S3 Table. Summary of included studies.**
(DOCX)

**S4 Table. PRISMA checklist.**
(DOCX)

## Author Contributions

**Conceptualization:** Nur Aishah Che Roos.

**Data curation:** Unwaniah Abdull Rahim.

**Formal analysis:** Unwaniah Abdull Rahim.

**Funding acquisition:** Nik Noorul Shakira Mohamed Shakrin, Armania Nurdin.

**Methodology:** Nur Aishah Che Roos.

**Project administration:** Nur Aishah Che Roos.

**Supervision:** Yasmin Anum Mohd Yusof, Nur Aishah Che Roos.

**Validation:** Nik Noorul Shakira Mohamed Shakrin, Armania Nurdin, Nursiati Mohamad Taridi, Mariam Firdhaus Mad Nordin.

**Visualization:** Unwaniah Abdull Rahim, Nur Aishah Che Roos.

**Writing – original draft:** Unwaniah Abdull Rahim, Nur Aishah Che Roos.

**Writing – review & editing:** Marami Mustapa, Nik Noorul Shakira Mohamed Shakrin, Armania Nurdin, Nur Aishah Che Roos.

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
