## [Decision Letter · Decision Letter 0]

21 Aug 2024

PONE-D-24-27562Current Evidence and Future Direction on Evaluating the Anticancer Effects of Curcumin, Gingerols, and Shogaols in Cervical Cancer: A Systematic ReviewPLOS ONE

Dear Dr. Che Roos,

Thank you for submitting your manuscript to PLOS ONE. After careful consideration, we feel that it has merit but does not fully meet PLOS ONE’s publication criteria as it currently stands. Therefore, we invite you to submit a revised version of the manuscript that addresses the points raised during the review process. As you can see, both reviewers I feel did like the paper, but also agree on the ideas that much more analysis and explanation is needed on the quality of studies reviewed, their relevance and in vivo data inclusion. Please address these concerns, taking close note on what the reviewers suggest. Re. reviewer #2, they suggest adding several references that they include. They are all from the reviewer's own work, and most are not relevant to this study. You can decide to include any of them at your discretion. I would however, include citations of 1-2 review articles on the therapeutic uses of curcumin or gingerols, as this is relevant to your manuscript. I would be happy to consider publication of a revised manuscript after concurrence by the reviewers. Please submit your revised manuscript by Oct 05 2024 11:59PM. If you will need more time than this to complete your revisions, please reply to this message or contact the journal office at plosone@plos.org. Please include the following items when submitting your revised manuscript:A rebuttal letter that responds to each point raised by the academic editor and reviewer(s). You should upload this letter as a separate file labeled 'Response to Reviewers'.A marked-up copy of your manuscript that highlights changes made to the original version. You should upload this as a separate file labeled 'Revised Manuscript with Track Changes'.An unmarked version of your revised paper without tracked changes. You should upload this as a separate file labeled 'Manuscript'.

We look forward to receiving your revised manuscript.

Kind regards,

Joseph J Barchi

Academic Editor

PLOS ONE

Journal Requirements:

2. Acknowledgments Section: Move New Information to the Financial Disclosure:

Thank you for stating the following in the Acknowledgments Section of your manuscript: This work was supported by the Fundamental Research Grant Scheme (FRGS) Ministry of Higher Education (MoHE) Malaysia (FRGS/1/2021/SKK0/UPNM/02/2), National Defence University of Malaysia (NDUM), and Universiti Putra Malaysia (UPM).

Please remove any funding-related text from the manuscript and let us know how you would like to update your Funding Statement. Currently, your Funding Statement reads as follows: The publication of this work is supported financially by

 1. The Fundamental Research Grant Scheme (FRGS) Ministry of Higher Education (MoHE) Malaysia (FRGS/1/2021/SKK0/UPNM/02/2) - Dr Nik Noorul Shakira Mohammed Shakrin

2. National Defence University of Malaysia (NDUM) - Dr Nur Aishah Che Roos

3. Universiti Putra Malaysia (UPM) -Dr Armania Nurdin

The funders do not play any role in the study design, data collection and analysis, decision to publish, or preparation of the manuscript.

Reviewers' comments:

Reviewer's Responses to Questions

**Comments to the Author**

1. Is the manuscript technically sound, and do the data support the conclusions?

Reviewer #1: Partly

Reviewer #2: Yes

2. Has the statistical analysis been performed appropriately and rigorously? 

Reviewer #1: Yes

Reviewer #2: Yes

3. Have the authors made all data underlying the findings in their manuscript fully available?

Reviewer #1: Yes

Reviewer #2: Yes

4. Is the manuscript presented in an intelligible fashion and written in standard English?

Reviewer #1: Yes

Reviewer #2: Yes

5. Review Comments to the Author

Reviewer #1: This paper aims to conduct a systematic review and meta analysis of the effect of curcumin, gingeral and shogaol on cervical cancer.

The paper presents a lot of information, but it is very difficult to digest. Table 1 has one line per study. This table should be in the appendix. Instead, please summarize across studies as much as possible. Table 2 presents all the results. It is difficult to understand which results have the most support. Please think about how to make this table more useful.

1. The paper is very broad including in vivo and in vitro, human cell lines and animal populations, any preparation of the compounds of interest, and many different kinds of outcomes. Please justify including all these different kinds of studies in the same review.

2. What is the rationale for excluding studies of human participants?

3. 12 studies were excluded due to the full text not being obtainable. How many and what attempts were made to secure these articles? Please discuss this as a limitation and consider how results could have been impacted by exclusion of such a high proportion of studies.

4. What is interesting about Table 2, is that there seems to be no replication. Did any studies use similar enough exposures, similar enough outcomes and come up with similar results?

5. Some statements are made, without evidence presented. "... the highest concentration of these compounds are associated with the lowest cell proliferation in all tested cell lines" Where are these results presented?

6. Generally, more care should be taken to present results in a way that highlights where there has been replication.

7. Among the excluded studies are 116 which were review articles. How has this paper improved upon previous attempts at reviewing this literature?

The information is not meta analyzed, but this is the correct decision as there is a lot of heterogeneity in the studies considered.

Reviewer #2: Comments to the author:

The current study The paper evaluates the anticancer effects of curcumin, gingerols, and shogaols specifically in cervical cancer and concludes that these compunds have promising anticancer properties. It can be said that the purpose of this study and its content are attractive and interesting. It offers a good perspective for improving the treatment of cervical cancer with natural products. The article is well written and the design of the study and analyzes are done appropriately and correctly. However, to improve the quality of this study, some major points should be addressed and corrected, which are as follows:

Major points:

1. The discussion mentions that most studies included were in vitro, which might not fully represent in vivo conditions. Could you elaborate on how this limitation impacts the generalizability of the findings to clinical settings?

2. The methodological quality of the in vitro studies was assessed as low. How does this low methodological quality affect the reliability of the overall conclusions drawn from the review?

3. There any significant inconsistencies in study designs, such as variations in cell lines, exposure times, or concentrations of bioactive compounds, that could have influenced the results. How were these addressed in the review?

4. Given the challenges in determining an optimal IC50 due to varying cell lines, exposure durations, and assay conditions, how should future research standardize these parameters to enhance comparability?

5. The study may not have fully explored the potential synergistic effects of combining these compounds with each other or with other therapeutic agents.

6. Challenges may exist related to the isolation and purification of these compounds for study purposes. How might these challenges not affect the reproducibility and scalability of research findings?

7. Were there any potential biases in the selection or interpretation of studies that the authors may not have fully addressed? How might these biases impact the conclusions drawn in the review?

8. Given the potential for chemoresistant cervical cancer cells to develop, how might one be sure about these bioactive compounds' effectiveness in overcoming resistance mechanisms?

9. As mentioned in this article, curcumin is a natural nutrient with high potential, which is used as a strong anti-inflammatory substance. Please mention the wide use of curcumin in various diseases (infections, ageing, cancers, and autoimmunity) in the introduction of a discussion and please consider including the following references in this article (doi:10.1007/s11357-024-01092-5, doi: 10.1016/j.lfs.2021.119437, and doi: 10.1016/j.ejphar.2022.175267 ).

10. The use of emerging and powerful treatments based on immunotherapy, such as tuberculosis therapy, has also improved the survival of patients. In the introduction section, explain about this type of treatment in a few sentences and consider including the following references (10.1186/s12935-022-02778-6, doi: 10.1007/s12094-020-02490-9, and doi: 10.1186/s12935-023-02923-9 ).

11. Another important issue that must be addressed is the presence of inflammatory cytokines in the tumor microenvironment and their confrontation with the PPAR marker due to the existence of common signaling pathways (“Activation of PPARγ negatively influences the production of inflammatory cytokines such as tumor necrosis factor-alpha (TNFα), Interleukin (IL)-6, and IL-1β by macrophages”.). Regarding the contrast with this, explain and consider including the following references in “Cancer Cell Proliferation and Viability” or “Cancer Cell Apoptosis” sections (doi: 10.1007/978-3-030-80962-1_25-1 and doi: 10.1155/2008/961753).

12. The quality and resolution of all figures are very less, please increase their quality.

13. There are some grammatical errors that should be corrected throughout the manuscript.

6. PLOS authors have the option to publish the peer review history of their article (what does this mean?). If published, this will include your full peer review and any attached files.

Reviewer #1: No

Reviewer #2: No

---

## [Author Response · Author response to Decision Letter 0]

30 Oct 2024

We have revised our manuscript according to most of the reviewers’ comments. Edited text and table in the attached revised manuscript is visible as tracked changes under the All Markup mode of Microsoft Word. The tracked manuscript also shows minor corrections and renumbered tables after revision. The order of the authors has also been changed based on their contribution to the revised manuscript. Our point-by-point responses are given in blue-coloured font in the attached file named ‘Response to Reviewers’. A clean file of the manuscript is also attached.

---

## [Editor Report · Decision Letter 1]

8 Nov 2024

Current Evidence and Future Direction on Evaluating the Anticancer Effects of Curcumin, Gingerols, and Shogaols in Cervical Cancer: A Systematic Review

PONE-D-24-27562R1

Dear Dr. Che Roos,

We’re pleased to inform you that your manuscript has been judged scientifically suitable for publication and will be formally accepted for publication once it meets all outstanding technical requirements.

Kind regards,

Joseph J Barchi

Academic Editor

PLOS ONE
---

## [Editor Report · Acceptance letter]

13 Nov 2024

PONE-D-24-27562R1 

PLOS ONE

Dear Dr. Che Roos, 

I'm pleased to inform you that your manuscript has been deemed suitable for publication in PLOS ONE. Congratulations! Your manuscript is now being handed over to our production team.

Kind regards, 

on behalf of

Dr. Joseph J Barchi 

Academic Editor

PLOS ONE